# Underuse of Gastric Cancer Screening Services among Koreans with Type 2 Diabetes

**DOI:** 10.3390/healthcare11070927

**Published:** 2023-03-23

**Authors:** Kumban Walter Chuck, Seri Hong, Yunhwan Lee

**Affiliations:** 1Department of Medical Sciences, Ajou University, 206 World cup-ro, Yeongtong-gu, Suwon-si 16499, Republic of Korea; 2Department of Preventive Medicine and Public Health, Ajou University School of Medicine, 164-World cup-ro, Yeongtong-gu, Suwon-si 16499, Republic of Korea

**Keywords:** diabetes mellitus, gastric cancer, screening, Korea

## Abstract

We aimed to compare the gastric cancer screening rates between participants with diabetes and those without diabetes in the Korean population. The data of 4284 participants from the Korea National Health and Nutrition Examination Survey 2019 were used. Cancer-free people aged ≥40 years were included, and cancer screening rates in diabetes and non-diabetes participants were analyzed. Baseline characteristics and screening rates were calculated using weighted frequencies and multivariable regression at a 95% confidence interval in both groups. Screening for gastric cancer was significantly lower (odd ratio [OR]: 0.77, 95% CI: 0.64–0.95) in patients with diabetes than in those without diabetes. The odds of performing the recommended gastric cancer screening were also lower (OR: 0.72, 95% CI: 0.58–0.90) in participants with diabetes than in those without diabetes. After adjusting for socio-demographic factors, the multivariable logistics regression analysis also showed lower odds for gastric cancer screening participation in diabetic patients than in non-diabetes participants. Conclusively, people with diabetes were less likely to have ever had or been recommended screening compared with those without diabetes. Greater efforts need to be made by health specialists to increase the awareness and the need of long-term preventive care including gastric cancer screening in high-risk groups.

## 1. Introduction

Globally, gastric cancer has been considered as the fifth most common human malignancy, ranked as the fourth-leading cause of cancer related deaths [1]; however, type 2 diabetes and cancer have contributed to increased mortality rates according to recent studies [2], and despite efforts to curb its occurrence, their existence persists, as they share several risks. Recent studies conducted in China have proven that hypoglycemia has a high probability of developing into gastric cancer, leading to an increase in gastric cancer incidences and the related death toll due to the proliferation of two major genes: caspase3 (CASP3) and tumor protein P53 (TP53) [3]. Nonetheless, the biological ties linking the two are still not well understood [4]. Research conducted involving 215 countries from 2005 to 2020 has reported a worldwide increase in the incidence of diabetes cases, ranging from approximately 10.5% in 2021 to a probable rise of approximately 12.2% in 2045 among people aged 20–79 years if appropriate management is not provided [5]; however, this has been proven to be associated with many risk factors and lifestyle activities, including aging, obesity, physical inactivity, diet, alcohol consumption, and smoking [6]. This value is expected to increase in the coming years; even though the number of diabetes cases decreased in 2002, it still remains the cause of increased mortality in Asian countries [7]. Nayyar and his collaborators in 2007 enumerated in their research that the burden of diabetes should be handled based on several considerations, especially in high-risk groups, including the control of glucose levels using strategies geared towards the modification of lifestyles, diet, and exercise in order to reduce its complications [8].

In the Republic of Korea, cancer has been a life-threatening issue in recent years, likely due to aging and Western lifestyles [9]; the worse possible consequences may be expected if proper preventive measures are not implemented [10]. In 1999, the National Cancer Screening Program (NCSP) was launched by the Korean health authorities to provide free screening services for gastric, breast, and cervical cancers, and the scope and number of participants for screening have increased since 2004 [11].

Although several recent studies have compared cancer screening rates in patients with diabetes, several non-similarities have also been observed [12,13,14]. In addition, only limited studies have been conducted in Korea to examine the disparity in cancer screening among diabetic patients (high-risk group) compared with their non-diabetic counterparts using recent data (2019). Despite the fact that other cancer types, including liver cancer, are more prominent in patients with type 2 diabetes, caused by high fructose concentrations that induce a series of pro-inflammatory fibrinogens and variable cancerous pathways, leading to chronic diseases including liver cancer [15], gastric cancer was chosen for this study because in the Korean population, recent studies have proven that gastric cancer is ranked first, with the highest incidence of cases (about 29,493 in 2019) [16] in both male and female populations, and secondly, gastric cancer has been known to have a poor prognosis, with a 5 year survival rate of less than 20% for those at the advanced stage of cancer infection [17]. Obesity was not left behind, as it is known to be associated with insulin resistance, oxidative stress, inflammation, and hyperinsulinemia, because some patients with diabetes may also be obese [18], and it is most probable that the shared risk factors may partly explain the high risk of gastric cancer in patients with diabetes, added to the fact that diabetic patients use anti-diabetic drugs, including metformin, aspirin, statins, and antibiotics; however, more studies are still needed to consolidate their linkage [19,20,21]. In vitro studies have indicated that hyperglycemia and gastric cancer are correlated, as their increase in glucose levels may affect the development of cancer through β-catenin acetylation with increased Wnt signaling [22]. Nonetheless, Helicobacter pylori infections have been be noted due to increased salt intake, which, to some extent, leads to gastric cancer [23,24]. Whether gastric cancer is directly linked to people with diabetes due to their increase in salt intake as a result of loss of taste remains a topic for further research. 

This study primarily aimed to investigate the use of gastric cancer screening services among Koreans with type 2 diabetes and to investigate potential disparities in gastric cancer screening rates with respect to socio-demographic factors using the 2019 Korea National Health and Nutrition Examination Survey (KHNANES) data. The 2019 survey dataset were chosen for this study to appreciate how gastric cancer screening behaviors have been influenced due to coronavirus (COVID-19). This study will contribute to increase the knowledge and promote the importance of cancer screening activities among people with chronic conditions, including diabetes; raise cancer control policy awareness; and determine the various setbacks of cancer screening. 

## 2. Methods

Data for this study were obtained from the KNHANES 2019 survey website (http://knhanes.cdc.go.kr) (accessed on the 6 May 2022). KNHANES 2019 is available using the following link with consent: https://knhanes.kdca.go.kr/knhanes/sub03/sub03_02_05.do (accessed on the 6 May 2022). This is a nationwide survey that has been conducted since 1998 and was initially conducted on a triennial basis. Since 2007, the survey has been conducted annually by the Korean Center for Disease Control and Prevention to evaluate the health and nutritional status of Koreans. These data are nationally represented, including approximately 10,000 individuals collected each year, comprising their socio-economic status, health-related behavior, quality of life, healthcare utilization, anthropometric measures, biochemical and clinical profiles for non-communicable disease, and dietary intake, with the findings of a three-component survey including direct physical examination, clinical laboratory examinations, and personal interviews [25].

Participants with a history of gastric (54), missing information on gastric cancer screening (402), those with fasting blood glucose testing of less than 8 hours (209), missing demographic data and missing data on diabetes diagnosis and treatment (943), and people of ages less than 40 years (2216) were excluded from the study population. Even though in recent studies, young people (aged less 30 years old) are prone to gastric cancer, they have also proven to have a varying prognosis [26]; however, in the Korean population, gastric cancer screening is recommended in people aged 40 years and above following the National Cancer Screening Program (NCSP) guidelines [27]. The diabetic population included people who have been diagnosed with diabetes by a physician and received treatment such as insulin therapy and other medications, those with fasting blood sugar levels of ≥126mg/dl, and those with glycated hemoglobin (HbA1c) levels of ≥6.5% [28]. The non-diabetic population included participants who did not meet the above criteria.

In the Korean population, the recommended gastric cancer screening is available for men and women 40 years old and above, and information about cancer screening, including socio-demographic characteristics, was collected by conducting a questionnaire survey. Information for this study was limited to men and women aged 40 years and above stratified in different age groups: 40–49, 50–59, 60–69, and 70 years and above. Income was analyzed based on the monthly household income: lowest (<1.0 million KRW per month), lower (<1.0–2.5 million KRW per month), higher (2.5–4.0 million KRW per month), and highest (>4.0 million KRW per month) (note: KRW 1300 = US $1). Educational levels were classified as elementary school, middle school, high school, and college or higher. The participants were classified by residential area: cities and districts including multiple “dong” (urban areas), whereas counties include multiple “eup” and “myeon” (rural areas). Medical insurance was divided into the National Health Insurance (NHI) program, which is compulsory for all residents of the Korean territory, and the Medical Aid Program (MAP), implemented by the government to enable minimum living standards for low-income households. The questions that were analyzed included “have you ever undergone gastric cancer screening?” If yes, the participants were asked about their screening method used and the last time they underwent screening. The cancer screening rate was estimated based on the number of screened patients, including those who underwent at least one screening in their lifetime (ever screening) and those who had endoscopy screening once every 2 years or an upper gastrointestinal series requested by a physician (recommended screening), in adults aged 40 to 74 years old according to the NCSP guidelines.

In addition to descriptive statistical analysis, the cancer screening rates according to study variables were stratified according to diabetes status and calculated along with a 95% confidence interval. The baseline characteristics of people with and without diabetes were calculated using weighted frequencies and compared using the chi-squared test. The weighted frequencies and screening rates were calculated as percentages of gastric cancer screening along with the 95% CI using summary statistics and standardized by age, and the screening rates were compared between people with diabetes and those without diabetes. Multivariable logistics regression was performed in this study to assess the screening rates and odds with a 95% CI for receiving gastric cancer screening in participants with diabetes and those without diabetes after adjusting for age, gender, income, education, residential area, and medical insurance. STATA version 12.0 (STATA Corp., College Station, TX, USA) was used to perform all statistical analyses.

## 3. Results

A total of 8110 participants were considered eligible for the study and those who could not meet the inclusion criteria were excluded. Finally, 4284 participants were included in the final analysis, as seen in Figure 1.

Table 1 shows the socio-demographic factors of the study participants, including age, gender, income, education, residential area, and type of health insurance. This table shows a total of 4284 participants, of which 706 were diabetic and 3518 had no diabetes. This table shows that there is a significant difference between participants with diabetes and those without diabetes. Overall, the rate of participation in gastric cancer screening was significantly higher in men aged ≥70 years, those less educated, and those with lower income levels than in those without diabetes. Individuals who simultaneously satisfied these two conditions (living in an urban area with NHI) with diabetes were less likely to be represented compared with those without diabetes.

Table 2 shows a comparison of the gastric cancer screening rates between people with diabetes and those without diabetes. Many participants had received gastric cancer screening (2966) compared to those who had been recommended gastric cancer screening (1942). There was a confounding effect of age observed, as diabetic participants aged 49–69 years were significantly less likely to have received screening ever and recommended gastric cancer screening compared to other aged groups. The ever screening rate was significantly lower (*p* < 0.001) in people with diabetes than in those without diabetes; the recommended cancer screening rate was lower (*p* < 0.009) in participants with diabetes than in those without diabetes. A subgroup analysis was conducted to see who underwent the recommended gastric cancer screening by age groups, gender, income, education, residential area, and the type of health security status; the observed recommended gastric cancer screening rates were lower in women aged 60 and younger, those living in urban areas, those with NHI, and those who satisfy these two conditions: highest income status and more than high school level of education in those with diabetes compared with those without diabetes. Women aged 70 years or younger, those living in urban areas, those with lower income status, those who completed high school or higher level, those who used NHI programs as health security, and those with diabetes showed significantly lower ever screening rates compared with those without diabetes.

Table 3 shows the results of the multivariable logistic regression analysis of the recommended and ever screening gastric cancer screening rates after adjusting for socio-demographic factors, including income levels, gender, age, educational levels, residential area, and health security status. People with diabetes showed significantly lower odds of undergoing the recommended gastric cancer screening (OR = 0.770, *p* < 0.016) compared with participants without diabetes. Participants with diabetes showed lower ever gastric cancer screening rates compared with those without diabetes (OR = 0.72, *p* < 0.004). With regards to the other socio-demographic factors, women aged ≥60 years showed a higher recommended cancer screening rate compared to those from other age groups. Participants with highest income levels and a lower than high school level of education also showed higher recommended gastric cancer screening rates compared with those with other income and educational statuses. Women aged ≥60 years showed higher ever gastric cancer screening rates compared with other age groups. Participants with higher income levels, with more than a middle school level of education, and those living in rural areas showed higher ever gastric cancer screening rates compared to those in other subgroups. Participants using the MAP as a means of health security showed lower ever screening rates compared with those using the NHI.

Figure 2 and Figure 3 show a graphical representation of gastric cancer screening trends (recommended and ever screening trends, respectively) with respect to diabetes status and age. In Figure 1, even though there was a steady increase in recommended cancer screening rates as age increases, participants with diabetes showed lower recommended cancer screening rates as compared to participants without diabetes. In Figure 2, even though there was a sharp decrease in ever screening rates in both diabetic and non-diabetic participants aged 60 years and above, there was an overall decrease in screening rates among participants with diabetes compared to those without diabetes. 

## 4. Discussion

This was the first study to use the recent KNHANES 2019 data to evaluate the underuse of gastric cancer screening rates in diabetic and non-diabetic participants in the Korean population. The results of this study reveal that people with diabetes had relatively lower ever and recommended gastric cancer screening rates compared with those without diabetes based on the results of the nationwide health and examination survey, even after launching the NCSP. Similar case control studies have recently shown that the presence of common chronic health problems, including diabetes, is strictly associated with reduced screening rates in the adult population, which agrees with the results of our study. These findings suggest that physicians need to ensure that patients receive all the recommended prevention services based on their age [29,30].

The results among the subgroups differed depending on their corresponding category: the confidence interval (CI) and *p*-values could not be obtained for all subgroups due to the uneven distribution of participants who were classified as “non- diabetes”. A few diabetic participants were accidentally included in this group, as they were unaware of their condition, and most of them belonged to health-vulnerable groups (older age, low income, and low education level). Participants in these subgroups have less interest in determining their health status, causing the screening rate to be relatively low. By contrast, even if they belong to a vulnerable group, people who are interested in recognizing their own health status understand that they are diabetic, and it is interpreted that people classified in the diabetes group will also have a relatively high screening rate. 

Since the prevalence of diabetes has continuously increased worldwide [31] and in Korea, [32] men and women with diabetes will remain at a higher risk of developing various types of cancers, including gastric cancer, with an increase in mortality rate [33]. Therefore, people with diabetes aged ≥40 years should undergo screening for various cancer types.

This study showed that women with diabetes showed reduced ever and recommended gastric cancer screening rates compared with those without diabetes, indicating that women were less vulnerable to screening, which was in agreement with the findings of recent studies conducted in the United States [31].

In Korea, invitation letters are sent to eligible populations for each type of cancer screening. Although those with diabetes have a higher probability of visiting the hospital to undergo glucose tests, their screening rates remained lower. This may be due to physicians focusing on the clinical management of diabetes and its complications and considering long-term management to be less important [13,30,34]. Furthermore, the screening rates for diabetes complications were also reduced [35], and previous studies stipulated that this may be due to increased time constraints perceived during screening [13,14,36]. Other studies have proven that this may be due to physicians being reluctant to provide preventive care to patients with chronic conditions, including diabetes, and preferring to focus on managing complicated cases [37]. However, because the incidence of diabetes has increased in recent years, diabetes should be managed using preventive health strategies, including cancer screening.

This study also showed lower screening rates among participants in their 70s with diabetes than among those without diabetes. This finding was in agreement with that of a recent study [38], suggesting that diabetic patients did not believe in the benefits of undergoing cancer screening for their survival. The level of education also showed discrepancies in gastric cancer screening rates between the study groups. Previous studies on socio-economic levels and educational levels indicated that education is directly related to the awareness on the population on the importance of screening, which may have greatly impacted the early detection rate and, thus, ameliorated the follow-up of cancer and its complications [12,36,39].

In Korea, the existence of the NCSP is thought to influence the cancer screening rates of citizens; however, high-income earners with diabetes still have lower screening rates compared to their non-diabetes counterparts. This finding was not different from the results of a study conducted in Korea on various cancers using the Korean National Cancer Screening Survey, which showed higher cancer screening rates in participants with higher income compared to those with lower income (<US $1500 dollars) per month. However, this study focused on individuals who underwent screening earlier [40] and emphasized that, despite the universal coverage of cancer screening in Korea, some political interference may lead to a reduction in the screening rates among people with lower income; hence, this issue should be carefully considered.

With regards to urbanity, this study showed significantly lower ever and recommended screening rates in participants with diabetes living in urban areas. This situation remains challenging, as Korea provides universal coverage for all citizens; hence, urbanity is not expected to cause any barrier with respect to access to cancer screening facilities. Nonetheless, rural areas have limited access to these services due to inadequate infra-structure and other related rural hindrances to healthcare facilities. Previous studies conducted abroad showed lower cancer screening rates in rural areas compared to urban areas due to socio economic disparity [41,42,43]. However, this could not be compared to the situation in Korea, as it has a NCSP. Participants with Medical Aid Programs as a source of social security showed relatively lower screening rates than those using NHI. This is obvious, as observed, because the NHI has a universal coverage, which offers many health benefits in contrast to MAP and provides health benefits to low-income earners who may have poor health status with limited coverage [44].

Lower gastric cancer screening rates observed in the older age groups may have been influenced by the lead and length time effects; the cause of this may be that this group may not have appreciated the advantages or benefits of cancer screening, as they believe it will not affect their survival outcomes. Research conducted in the Korean population on gastric cancer screening according to stages showed that even though these effects (length and lead time) were thought to increase the survival of the screened population, in reality, this had just helped shorten the diagnosis period, thus extending the period between diagnosis and death without prolonging their life [45].

This study has several limitations. First, we included self-reported data on the diabetes status and the receiver of cancer screenings, which are liable to recall bias. In addition, self-reported data were used without matching with medical records, which may lead to information bias. We were unable to differentiate the various types of diabetes (type 1 and 2) and their duration, which could affect cancer risk differently based on the screening guidelines. Moreover, we did not have enough information to measure the efficacy of diabetes care management programs. Lastly, the diabetes population was small due to a reduced sample size; however, future research will necessitate the combination of different datasets.

## 5. Conclusions

As people with diabetes continue to be vulnerable, their underuse of preventive care services, including cancer screening, remains a point of concern to public health professionals. This study showed lower gastric cancer screening rates in people with diabetes compared with those without diabetes. These findings were in line with the reports of similar studies, although they used different survey years. Thus, our findings propose that public health practitioners should educate people with chronic diseases, including diabetes, on the importance of undergoing cancer screening and adopting good health practices. Future research should focus on investigating the different cancer types and their relationships to type 2 diabetes, the need for screening per diabetes duration, and the types of medication used.

## Figures and Tables

**Figure 1 healthcare-11-00927-f001:**
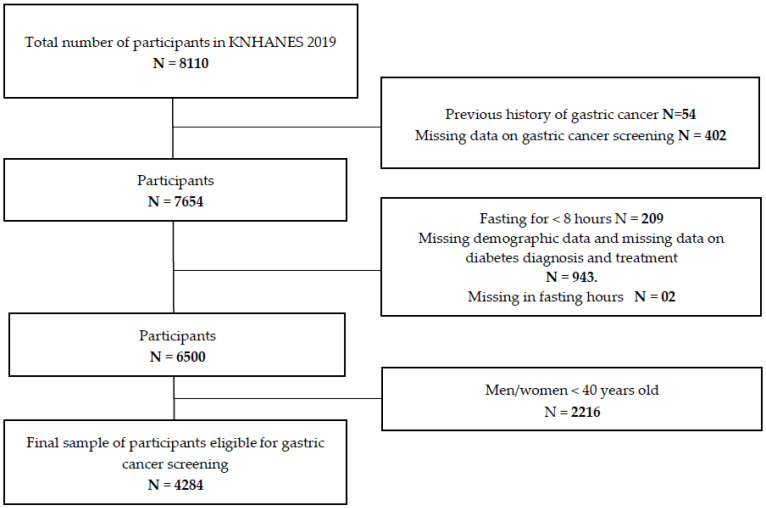
Flow chart describing gastric cancer screening sample selection.

**Figure 2 healthcare-11-00927-f002:**
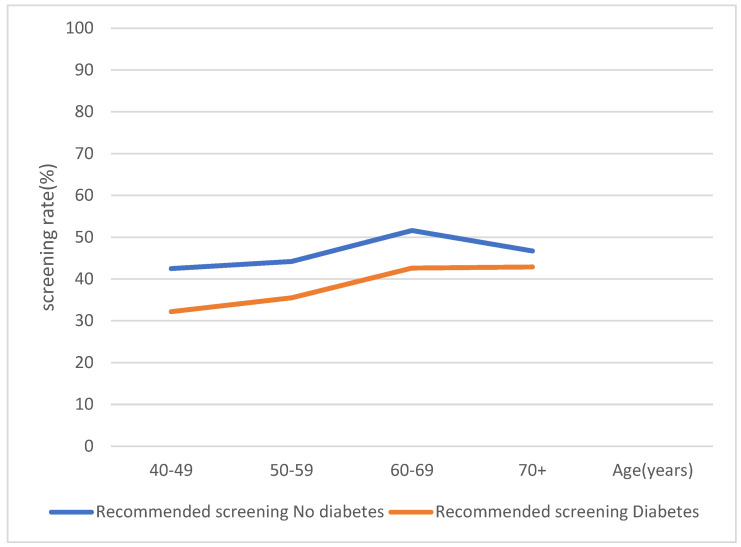
Recommended gastric cancer screening according to diabetes status and age.

**Figure 3 healthcare-11-00927-f003:**
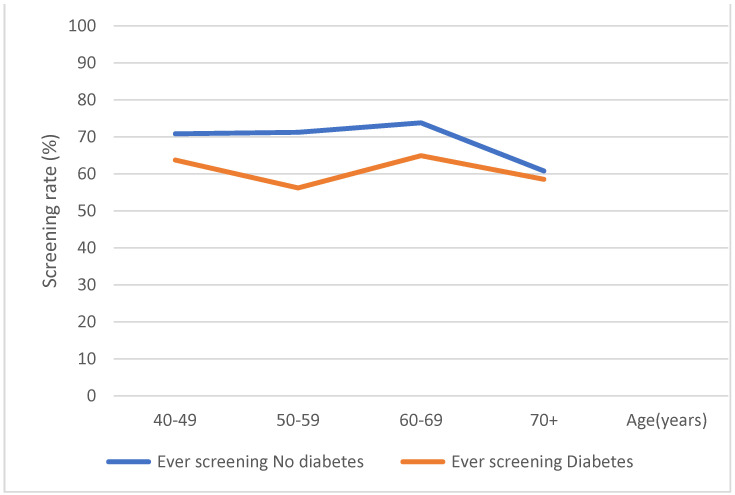
Ever gastric cancer screening rate according to diabetes status and age.

**Table 1 healthcare-11-00927-t001:** Socio-demographic factors of the study participants.

	Total (N (%))	Diabetes (N (%))	Without Diabetes (N (%))	*p*-Value
Age (years)	4284 (100)	706 (16.48)	3578 (83.52)	
40–49	1086 (25.35)	72 (10.20)	1014 (28.34)	
50–59	1132 (26.42)	138 (19.55)	994 (27.78)	<0.001
60–69	1055 (24.64)	229 (32.44)	826 (23.09)	
70+	1011 (23.60)	267 (37.82)	744 (20.79)	
Gender				
Men	1825 (42.60)	372 (52.69)	1453 (40.61)	
Women	2459 (57.40)	334 (47.31)	2125 (59.39)	<0.001
Income				
Lowest	971 (22.78)	230 (32.67)	741 (20.82)	
Lower	1109 (26.01)	197 (27.98)	912 (25.63)	
Higher	1034 (24.26)	139 (19.74)	895 (25.15)	<0.001
Highest	1149 (26.95)	138 (19.60)	1011 (28.41)	
Education				
Elementary school	958 (23.62)	245 (37.12)	713 (21.00)	
Middle school	508 (12.52)	109 (16.52)	399 (11.75)	<0.001
High school	1328 (32.74)	187 (28.33)	1141 (33.60)	
College and more	1262 (31.11)	119 (18.03)	1143 (33.66)	
Residential area				
Urban	3367 (78.59)	526 (74.50)	2841 (79.40)	<0.004
Rural	917 (21.41)	180 (25.50)	737 (20.60)	
Health insurance				
NHI	4097 (95.6)	656 (92.92)	3441 (96.20)	
MAP	186 (4.34)	50 (7.08)	136 (3.80)	<0.001

Values are presented as weighted frequencies and percentages (%). NHI, National Health Insurance; MAP, Medical Aid Program.

**Table 2 healthcare-11-00927-t002:** Ever and recommended gastric cancer screening rates in diabetes and non-diabetes participants.

	Recommended Screening 1942 (45.33%)			Ever Screening 2966 (69.23%)		
Total	Diabetes	No Diabetes	*p*-Value	Diabetes	No Diabetes	*p*-Value
	(95% CI)	(95% CI)		(95% CI)	(95% CI)	
Total	39.4 (35.3–43.6)	45.5 (43.6–47.3)	<0.009	60.5 (56.2–64.6)	70.0 (68.2–71.7)	<0.001
Age (years)						
40–49	32.2 (X)	42.5 (X)	/	63.7 (X)	70.8 (X)	/
50–59	35.5 (27.3–44.6)	44.2 (40.8–47.7)	<0.039	56.2 (46.7–65.2)	71.2 (67.8–74.3)	<0.001
60–69	42.6 (35.8–49.7)	51.6 (47.8–55.5)	<0.029	64.9 (57.5–71.6)	73.8 (70.1–77.1)	<0.021
70+	42.9 (36.4–49.7)	46.7 (42.7–50.8)	<0.349	58.5 (51.5–65.1)	60.8 (56.8–64.7)	<0.559
Gender						
Men	38.5 (33.1–44.36)	39.7 (36.9–42.6)	<0.707	62.7 (56.7–68.2)	67.0 (64.2–69.7)	<0.178
Women	40.5 (34.6–46.5.)	50.5 (48.2–52.8)	<0.002	57.6 (51.4–63.7)	72.6 (70.4–74.8)	<0.001
Income						
Lowest	38.0 (31.3–54.2)	44.2 (40.2–48.2)	<0.144	54.2 (46.8–61.5)	59.1 (54.8–63.0)	<0.262
Lower	41.2 (33.6–49.2)	45.6 (41.8–49.3)	<0.333	55.7 (47.4–63.7)	66.7 (62.9–70.3)	<0.013
Higher	42.2 (33.3–51.5)	42.8 (39.3–46.5)	<0.887	59.3 (49.5–68.3)	68.3 (64.6–71.7)	<0.071
Highest	36.6 (27.9–46.3)	48.5 (45.1–52.0)	<0.022	74.1 (65.0–81.4)	80.6 (77.6–83.2)	<0.115
Education						
Elementary school	42.5 (35.8–49.5)	49.3 (45.1–53.5)	<0.102	59.3 (52.1–65.9)	66.4 (62.2–70.3)	<0.076
Middle school	48.6 (33.5–55.2)	53.9 (41.3–62.5)	<0.081	69.5 (55.2–75.1)	75.1 (65.8–81.5)	<0.621
High School	40.8 (33.1–49.1)	48.9 (45.7–52.1)	<0.054	61.9 (53.2–69.8)	73.5 (70.5–76.4)	<0.006
College and more	38.7 (25.7–46.7)	44.4 (32.5–67.8)	<0.004	73.5 (60.1–80.2)	76.3 (57.0–87.4)	<0.004
Residential area						
Urban	39.2 (34.5–44.1)	45.6 (43.5–47.6)	<0.018	58.5 (53.5–63.3)	70.2 (68.2–72.1)	<0.001
Rural	39.9 (32.2–48.2)	45.1 (41.0–49.1)	<0.278	67.6 (59.1–75.2)	69.1 (65.1–72.8)	<0.751
Health Insurance						
NHI	39.5 (35.3–43.5)	45.7 (43.8–47.6)	<0.009	61.8 (57.3–66.1)	70.9 (69.1–72.6)	<0.001
MAP	37.6(X)	36.9(X)	/	38.9(X)	43.7(X)	/

**Table 3 healthcare-11-00927-t003:** Multivariable logistic regression analysis of the diabetes status and socio-demographic factors.

	Recommended Screening		Ever Screening	
	OR (95%CI)	*p*-Value	OR (95%CI)	*p*-Value
Diabetes				
No	Ref		Ref	
Yes	0.77 (0.64–0.95)	<0.016	0.72 (0.58–0.90)	<0.004
Age (years)				
40–49	Ref		Ref	
50–59	1.04 (0.85–1.27)	<0.681	0.97 (0.77–1.23)	<0.857
60–69	1.48 (1.18–1.87)	<0.001	1.49 (1.13–1.96)	<0.005
70+	1.62 (1.24–2.09)	<0.001	1.35 (1.00–1.81)	<0.048
Gender				
Men	Ref		Ref	
Women	1.48 (1.27–1.72)	<0.001	1.33 (1.12–1.58)	<0.001
Income				
Lowest	Ref		Ref	
Lower	1.15 (0.92–1.45)	<0.209	1.19 (0.94–1.53)	<0.153
Higher	1.11 (0.87–1.42)	<0.387	1.32 (1.02–1.73)	<0.038
Highest	1.33 (1.04–1.71)	<0.026	2.46 (1.83–3.31)	<0.001
Education				
Elementary school	Ref		Ref	
Middle school	1.43 (1.09–1.86)	<0.008	1.56 (1.16–2.10)	<0.003
High School	1.27 (1.01–1.61)	<0.045	1.39 (1.07–1.81)	<0.013
College and more	1.13 (0.87–1.48)	<0.341	1.57 (1.17–2.12)	<0.003
Residential area				
Urban	Ref		Ref	
Rural	1.01 (0.58–1.29)	<0.879	1.24 (1.01–1.53)	<0.042
Health Insurance				
NHI	Ref		Ref	
MAP	0.86 (0.58–1.21)	<0.487	0.50 (0.35–0.76)	<0.001

Ref, reference; CI, confidence interval; NHI, National Health Insurance; MAP, Medical Aid Program; OR, odds ratio.

## Data Availability

Data sharing not applicable.

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
