# Peer review of "Underuse of Gastric Cancer Screening Services among Koreans with Type 2 Diabetes"

_healthcare, 2023, doi:10.3390/healthcare11070927_

Round 1

Reviewer 1 Report

Comments to the Author

The authors conducted a retrospective observational study to investigate the difference in gastric cancer participation rate between diabetic and non-diabetic patients. Although the manuscript is written comprehensively, some concerns should be addressed.

Major

1.     Methods. Please describe the way of statistical analysis more in detail. What kind of statistical test did the authors use?

2.     Results. How many participants were classified into ‘Recommended screening’ or ‘ Ever screening’?

3.     Results. What do the authors think about the confounding effect of age on the relationship between diabetes and screening conditions?

Reviewer 2 Report

Dear Author

The manuscript entitled" Under Use of Gastric Cancer Screening Services Among Koreans with Type 2 Diabetes by Walter Chuck et al  is an interesting research article which provides the awareness on the screening the gastric cancer in type 2 diabetes patients among Korean population. In this research article the authors aimed to compare the gastric cancer screening rate between participants with and without diabetes. Through his research findings using data available from NHANES the authors found that the gastric screening rate was significantly lower in people with diabetes than non-diabetic people through various bioinformatic tools. Even though the research article looks interesting in which the authors compared the various socioeconomical tools for the investigation but the authors misses some key information and need some clarifications regarding the manuscript. The authors need to improve the manuscript in a way with some additional databases related to diabetes and gastric cancer will be more convincing and interesting to the audience. These are the major concerns the authors need to response in order to improve the quality of the manuscript.

1.      The authors need to explain what is the main reason for excluding the patients less than 40 years old. Since in the recent years diabetic is also more prominently found in the teenage people also and why particularly the authors concentrate on gastric cancer rather than other major cancer in which diabetes is more likely linked. For example, increased intake of high fructose diet leads to type 2 diabetes and more likely liver cancer.

2.      The authors need to provide the datas in the interesting way in which pay attention towards the audience. Did authors used any public databases and try to connect the gastric cancer with people with and without diabetes and represented in a graphical representation.

3.      The authors usage of the sample size of diabetic patients was found to be very less than non-diabetic since the manuscript focused on the connection between type 2 diabetes and gastric cancer.

4.      The authors need to explain how type 2 diabetes and gastric cancer can be linked in order to get visibility of the article. In addition, the authors need to mention what are all the common gene expression elevated in both diabetic and gastric cancer patients so which the screening rate can be increased.
